# Convergence of Alternating Gradient Descent for Matrix Factorization

**Rachel Ward**
University of Texas
Austin, TX
rward@math.utexas.edu

**Tamara G. Kolda**
MathSci.ai
Dublin, CA
tammy.kolda@mathsci.ai

## Abstract

We consider alternating gradient descent (AGD) with fixed step size applied to the asymmetric matrix factorization objective. We show that, for a rank-$r$ matrix $\mathbf{A} \in \mathbb{R}^{m \times n}$, $T = C\left(\frac{\sigma_1(\mathbf{A})}{\sigma_r(\mathbf{A})}\right)^2 \log(1/\epsilon)$ iterations of alternating gradient descent suffice to reach an $\epsilon$-optimal factorization $\|\mathbf{A} - \mathbf{X}_T\mathbf{Y}_T^\mathsf{T}\|_\mathrm{F}^2 \leq \epsilon \|\mathbf{A}\|_\mathrm{F}^2$ with high probability starting from an atypical random initialization. The factors have rank $d \geq r$ so that $\mathbf{X}_T \in \mathbb{R}^{m \times d}$ and $\mathbf{Y}_T \in \mathbb{R}^{n \times d}$, and mild overparameterization suffices for the constant $C$ in the iteration complexity $T$ to be an absolute constant. Experiments suggest that our proposed initialization is not merely of theoretical benefit, but rather significantly improves the convergence rate of gradient descent in practice. Our proof is conceptually simple: a uniform Polyak-Łojasiewicz (PL) inequality and uniform Lipschitz smoothness constant are guaranteed for a sufficient number of iterations, starting from our random initialization. Our proof method should be useful for extending and simplifying convergence analyses for a broader class of nonconvex low-rank factorization problems.

## 1 Introduction

This paper focuses on the convergence behavior of alternating gradient descent (AGD) on the low-rank matrix factorization objective

$$\min f(\mathbf{X}, \mathbf{Y}) \equiv \frac{1}{2}\|\mathbf{X}\mathbf{Y}^\mathsf{T} - \mathbf{A}\|_\mathrm{F}^2 \quad \text{subject to} \quad \mathbf{X} \in \mathbb{R}^{m \times d}, \mathbf{Y} \in \mathbb{R}^{n \times d}. \tag{1}$$

Here, we assume $m, n \gg d \geq r = \text{rank}(\mathbf{A})$. While there are a multitude of more efficient algorithms for low-rank matrix approximation, this serves as a simple prototype and special case of more complicated nonlinear optimization problems where gradient descent (or stochastic gradient descent) is the method of choice but not well-understood theoretically. Such problems include low-rank tensor factorization using the GCP algorithm descent [HKD20], a stochastic gradient variant of the GCP algorithm [KH20], as well as deep learning optimization.

Surprisingly, the convergence behavior of gradient descent for low-rank matrix factorization is still not completely understood, in the sense that there is a large gap between theoretical guarantees and empirical performance. We take a step in closing this gap, providing a sharp linear convergence rate from a simple asymmetric random initialization. Precisely, we show that if $\mathbf{A}$ is rank-$r$, then a number of iterations $T = C\frac{d}{(\sqrt{d}-\sqrt{r-1})^2}\frac{\sigma_1^2(\mathbf{A})}{\sigma_r^2(\mathbf{A})}\log(1/\epsilon)$ suffices to obtain an $\epsilon$-optimal factorization with high probability. Here, $\sigma_k(\mathbf{A})$ denotes the $k$th singular value of $\mathbf{A}$ and $C > 0$ is a numerical constant. To the authors' knowledge, this improves on the state-of-art convergence result in the

37th Conference on Neural Information Processing Systems (NeurIPS 2023).

literature [JCD22], which provides an iteration complexity $T = C\left(\left(\frac{\sigma_1(\mathbf{A})}{\sigma_r(\mathbf{A})}\right)^3 \log(1/\epsilon)\right)$ for gradient descent to reach an $\epsilon$-approximate rank-$r$ approximation[1].

Our improved convergence analysis is facilitated by our choice of initialization of $\mathbf{X}_0, \mathbf{Y}_0$, which appears to be new in the literature and is distinct from the standard Gaussian initialization. Specifically, for $\mathbf{\Phi}_1$ and $\mathbf{\Phi}_2$ independent Gaussian matrices, we consider an "unbalanced" random initialization of the form $\mathbf{X}_0 \sim \frac{1}{\sqrt{\eta}}\mathbf{A}\mathbf{\Phi}_1$ and $\mathbf{Y}_0 \sim \sqrt{\eta}\mathbf{\Phi}_2$, where $\eta > 0$ is the step-size used in (alternating) gradient descent. A crucial feature of this initialization is that the columns of $\mathbf{X}_0$ are in the column span of $\mathbf{A}$, and thus by invariance of the alternating gradient update steps, the columns of $\mathbf{X}_t$ remain in the column span of $\mathbf{A}$ throughout the optimization. Because of this, a positive $r$th singular value of $\mathbf{X}_t$ provides a Polyak-Łojasiewicz (PL) inequality for the region of the loss landscape on which the trajectory of alternating gradient descent is guaranteed to be confined to, even though the matrix factorization loss function $f$ in (1) does not satisfy a PL-inequality globally.

By Gaussian concentration, the pseudo-condition numbers $\frac{\sigma_1(\mathbf{X}_0)}{\sigma_r(\mathbf{X}_0)} \sim \frac{\sigma_1(\mathbf{A})}{\sigma_r(\mathbf{A})}$ are comparable with high probability[2]; for a range of step-size $\eta$ and the unbalanced initialization $\mathbf{X}_0 \sim \frac{1}{\sqrt{\eta}}\mathbf{A}\mathbf{\Phi}_1$ and $\mathbf{Y}_0 \sim \sqrt{\eta}\mathbf{\Phi}_2$, we show that $\frac{\sigma_1(\mathbf{X}_t)}{\sigma_r(\mathbf{X}_t)}$ is guaranteed to remain comparable to $\frac{\sigma_1(\mathbf{A})}{\sigma_r(\mathbf{A})}$ for a sufficiently large number of iterations $t$ that we are guaranteed a linear rate of convergence with rate $\left(\frac{\sigma_r(\mathbf{X}_0)}{\sigma_1(\mathbf{X}_0)}\right)^2$.

The unbalanced initialization, with the particular re-scaling of $\mathbf{X}_0$ and $\mathbf{Y}_0$ by $\frac{1}{\sqrt{\eta}}$ and $\sqrt{\eta}$, respectively, is not a theoretical artifact but crucial in practice for achieving a faster convergence rate compared to a standard Gaussian initialization, as illustrated in Fig. 1. Also in Fig. 1, we compare empirical convergence rates to the theoretical rates derived in Theorem 3.1 below, indicating that our rates are sharp and made possible only by our particular choice of initialization. The derived convergence rate of (alternating) gradient descent starting from the particular asymmetric initialization where the columns of $\mathbf{X}_0$ are in the column span of $\mathbf{A}$ can be explained intuitively as follows: in this regime, the $\mathbf{X}_t$ updates remain sufficiently small with respect to the initial scale of $\mathbf{X}_0$, while the $\mathbf{Y}_t$ updates change sufficiently quickly with respect to the initial scale $\mathbf{Y}_0$, that the resulting alternating gradient descent dynamics on matrix factorization follow the dynamics of gradient descent on the linear regression problem $\min g(\mathbf{Y}) = \|\mathbf{X}_0\mathbf{Y}^T - \mathbf{A}\|_F^2$ where $\mathbf{X}_0$ is held fixed at its initialization.

We acknowledge that our unbalanced initialization of $\mathbf{X}_0$ and $\mathbf{Y}_0$ is different from the standard Gaussian random initialization in neural network training, which is a leading motivation for studying gradient descent as an algorithm for matrix factorization. The unbalanced initialization should not be viewed as at odds with the implicit bias of gradient descent towards a balanced factorization [CKL+21, WCZT21, ABC+22, CB22], which have been linked to better generalization performance in various neural network settings. An interesting direction of future research is to compare the properties of the factorizations obtained by (alternating) gradient descent, starting from various (balanced versus unbalanced) initializations.

## 2 Preliminaries

Throughout, for an $m \times n$ matrix $\mathbf{M}$, $\|\mathbf{M}\|$ refers to the spectral norm and $\|\mathbf{M}\|_F$ refers to the Frobenius norm.

Consider the square loss applied to the matrix factorization problem (1). The gradients are

$$\nabla_{\mathrm{x}} f(\mathbf{X}, \mathbf{Y}) = (\mathbf{X}\mathbf{Y}^\intercal - \mathbf{A})\mathbf{Y}, \tag{2a}$$
$$\nabla_{\mathrm{y}} f(\mathbf{X}, \mathbf{Y}) = (\mathbf{X}\mathbf{Y}^\intercal - \mathbf{A})^\intercal \mathbf{X}. \tag{2b}$$

We will analyze alternating gradient descent, defined as follows.

---

[1]We note that our results are not precisely directly comparable as our analysis is for alternating gradient descent whereas existing results hold for gradient descent. However, empirically, alternating and non-alternating gradient descent exhibit similar behavior across many experiments

[2]The pseudo-condition number, $\frac{\sigma_1(\mathbf{A})}{\sigma_r(\mathbf{A})}$, is equivalent to and sometimes discussed as the product of the product of the spectral norms of the matrix and its pseudoinverse, i.e., $\|\mathbf{A}\|\|\mathbf{A}^\dagger\|$

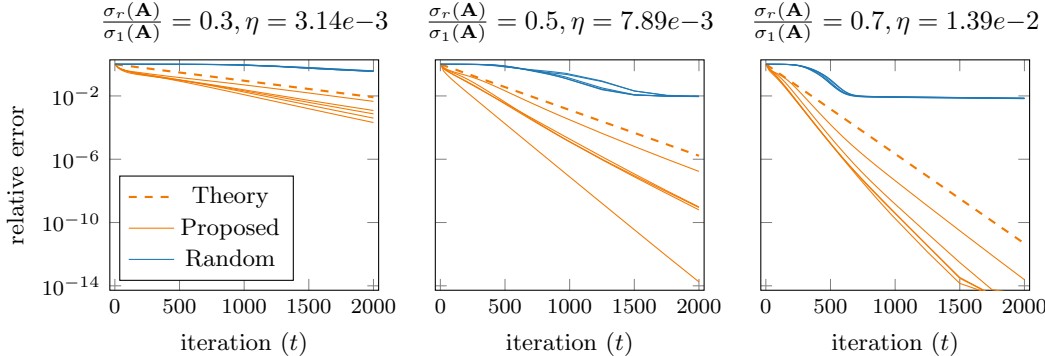

Figure 1: Alternating gradient descent for $\mathbf{A} \in \mathbb{R}^{100 \times 100}$ with $\mathrm{rank}(\mathbf{A}) = 5$ and factors of size $100 \times 10$, The plot shows five runs each of our proposed initialization and compared with the standard random initialization. The title of each plot shows the condition and step length.

---

*Assumption* 1 (Alternating Gradient Descent). For fixed stepsize $\eta > 0$ and initial condition $(\mathbf{X}_0, \mathbf{Y}_0)$, the update is

$$\mathbf{X}_{t+1} = \mathbf{X}_t - \eta \nabla_{\mathbf{x}} f(\mathbf{X}_t, \mathbf{Y}_t), \tag{A1a}$$

$$\mathbf{Y}_{t+1} = \mathbf{Y}_t - \eta \nabla_{\mathbf{Y}} f(\mathbf{X}_{t+1}, \mathbf{Y}_t). \tag{A1b}$$

---

We assume that the iterations are initialized in an asymmetric way, which depends on the step size $\eta$ and assumes a known upper bound on the spectral norm of $\mathbf{A}$. The matrix factorization is of rank $d > r$, and we also make assumptions about the relationship of $d$, $r$, and quantities $s$, $\beta$, and $\delta$ that will impact the bounds on the probability of finding and $\epsilon$-optimal factorization.

---

*Assumption* 2 (Initialization and key quantities). Draw random matrices $\mathbf{\Phi}_1, \mathbf{\Phi}_2 \in \mathbb{R}^{n \times d}$ with i.i.d. $\mathcal{N}(0, 1/d)$ and $\mathcal{N}(0, 1/n)$ entries, respectively. Fix $C \geq 1$, $\nu < 1$, and $D \leq \frac{C}{9}\nu$, and let

$$\mathbf{X}_0 = \frac{1}{\eta^{1/2} C \sigma_1(\mathbf{A})} \mathbf{A}\mathbf{\Phi}_1, \quad \text{and} \quad \mathbf{Y}_0 = \eta^{1/2} D \sigma_1(\mathbf{A}) \mathbf{\Phi}_2. \tag{A2a}$$

The factor matrices each have $d \geq r$ columns.
For $\tau > 0$, define

$$\rho = \tau \left(1 - \frac{\sqrt{r-1}}{\sqrt{d}}\right) \tag{A2b}$$

The number of iterations for convergence to $\epsilon$-optimal factorization will ultimately be shown to depend on

$$\beta = \frac{\rho^2 \sigma_r^2(\mathbf{A})}{C^2 \sigma_1^2(\mathbf{A})}. \tag{A2c}$$

The probability of finding this $\epsilon$-optimal factorization will depend on

$$\delta = (C_1 \tau)^{d-r+1} + e^{-C_2 d} + e^{-r/2} + e^{-d/2} \tag{A2d}$$

where $C_1, C_2 > 0$ are the universal constants in A.1.

---

Observe that the initialization of $\mathbf{X}_0$ ensures its columns are in the column span of $\mathbf{A}$.

---

*Remark* 2.1. The quantity $f(\mathbf{X}_0, \mathbf{Y}_0)$ does not depend on the step size $\eta$ or $\sigma_1(\mathbf{A})$ in Assumption 2 since

$$\mathbf{X}_0 \mathbf{Y}_0^\mathsf{T} - \mathbf{A} = \mathbf{A}\left(\frac{D}{C}\mathbf{\Phi}_1 \mathbf{\Phi}_2^\mathsf{T} - \mathbf{I}\right).$$

---

# 3 Main results

Our first main result gives a sharp guarantee on the number of iterations necessary for alternating gradient descent to be guaranteed to produce an $\epsilon$-optimal factorization.

---

**Theorem 3.1** (Main result, informal). *For a rank-$r$ matrix $\mathbf{A} \in \mathbb{R}^{m \times n}$, set $d \geq r$ and consider $\mathbf{X}_0, \mathbf{Y}_0$ randomly initialized as in Assumption 2. For any $\epsilon > 0$, there is an explicit step-size $\eta = \eta(\epsilon) > 0$ for alternating gradient descent as in Assumption 1 such that*

$$\|\mathbf{A} - \mathbf{X}_T \mathbf{Y}_T^\intercal\|_\mathrm{F}^2 \leq \epsilon \quad \textit{for all} \quad T \geq C \frac{\sigma_1^2(\mathbf{A})}{\sigma_r^2(\mathbf{A})} \frac{1}{\rho^2} \log \frac{\|\mathbf{A}\|_\mathrm{F}^2}{\epsilon}$$

*with probability $1 - \delta$ with respect to the draw of $\mathbf{X}_0$ and $\mathbf{Y}_0$ where $\delta$ is defined in (A2d). Here, $C > 0$ is an explicit numerical constant.*

---

For more complete theorem statements, see Corollary 5.2 and Corollary 5.3 below.

We highlight a few points below.

1. The iteration complexity in Theorem 3.1 is independent of the ambient dimensions $n, m$. In the edge case $d = r$, $\frac{1}{\rho^2} = O(r^2)$, so the iteration complexity scales quadratically with $r$. With mild multiplicative overparameterization $d = (1 + \alpha)r$, $\frac{1}{\rho^2} = \frac{(1+\alpha)}{(\sqrt{1+\alpha}-1)^2}$, and the iteration complexity is essentially dimension-free. This is a direct result of the dramatic improvement in the condition number of a $(1 + \alpha)r \times r$ Gaussian random matrix compared to the condition number of a square $r \times r$ Gaussian random matrix.

2. Experiments illustrate that initializing $\mathbf{X}_0$ in the column span of $\mathbf{A}$, and especially re-scaling $\mathbf{X}_0$ and $\mathbf{Y}_0$ by $\frac{1}{\sqrt{\eta}}$ and $\sqrt{\eta}$, respectively, is crucial in practice for improving the convergence rate of gradient descent. See Figs. 2 to 4.

3. The iteration complexity in Theorem 3.1 is conservative. In experiments, the convergence rate often follows a dependence on $\frac{\sigma_r(\mathbf{A})}{\sigma_1(\mathbf{A})}$ rather than $\frac{\sigma_r^2(\mathbf{A})}{\sigma_1^2(\mathbf{A})}$ for the first several iterations.

## 3.1 Our contribution and prior work

The seminal work of Burer and Monteiro [BM03, BM05] advocated for the general approach of using simple algorithms such as gradient descent directly applied to low-rank factor matrices for solving non-convex optimization problems with low-rank matrix solutions. Initial theoretical work on gradient descent for low-rank factorization problems such as [ZWL15], [TBS+16], [ZL16], [SWW17], [BKS16] did not prove global convergence of gradient descent, but rather local convergence of gradient descent starting from a spectral initialization (that is, an initialization involving SVD computations). In almost all cases, the spectral initialization is the dominant computation, and thus a more global convergence analysis for gradient descent is desirable.

Global convergence for gradient descent for matrix factorization problems without additional explicit regularization was first derived in the symmetric setting, where $\mathbf{A} \in \mathbb{R}^{n \times n}$ is positive semi-definite, and $f(\mathbf{X}) = \|\mathbf{A} - \mathbf{X}\mathbf{X}^\intercal\|_\mathrm{F}^2$, see for example [GHJY15, JJKN17, CCFM19].

For overparameterized symmetric matrix factorization, the convergence behavior and implicit bias towards particular solutions for gradient descent with small step-size and from small initialization was analyzed in the work [GWB+17, LMZ18, ACHL19, CGMR20].

The paper [YD21] initiated a study of gradient descent with fixed step-size in the more challenging setting of *asymmetric* matrix factorization, where $\mathbf{A} \in \mathbb{R}^{m \times n}$ is rank-$r$ and the objective is $\|\mathbf{A} - \mathbf{X}\mathbf{Y}^\intercal\|_\mathrm{F}^2$. This work improved on previous work in the setting of gradient flow and gradient descent with decreasing step-size [DHL18]. The paper [YD21] proved an iteration complexity of $T = \mathcal{O}\big(nd\big(\frac{\sigma_1(\mathbf{A})}{\sigma_1(\mathbf{A})}\big)^4 \log(1/\epsilon)\big)$ for reaching an $\epsilon$-approximate matrix factorization, starting from small i.i.d. Gaussian initialization for the factors $\mathbf{X}_0, \mathbf{Y}_0$. More recently, [JCD22] studied gradient descent for asymmetric matrix factorization, and proved an iteration complexity $T = \mathcal{O}\big(C_d\big(\frac{\sigma_1(\mathbf{A})}{\sigma_r(\mathbf{A})}\big)^3 \log(1/\epsilon)\big)$ to reach an $\epsilon$-optimal factorization, starting from small i.i.d. Gaussian initialization.

We improve on previous analysis of gradient descent applied to objectives of the form (1), providing an improved iteration complexity $T = \mathcal{O}\big(\big(\frac{\sigma_1(\mathbf{A})}{\sigma_r(\mathbf{A})}\big)^2 \log(1/\epsilon)\big)$ to reach an $\epsilon$-approximate factorization. There is no dependence on the matrix dimensions in our bound, and the dependence on the rank $r$ disappears if the optimization is mildly over-parameterized, i.e., $d = (1 + \alpha)r$. We do note that our results are not directly comparable to previous work as we analyze alternating gradient descent rather than full gradient descent. Our method of proof is conceptually simpler than previous works; in particular, because our initialization $\mathbf{X}_0$ is in the column span of $\mathbf{A}$, we do not require a two-stage analysis and instead can prove a fast linear convergence from the initial iteration.

## 4 Preliminary lemmas

---

**Lemma 4.1** (Bounding sum of norms of gradients). *Consider alternating gradient descent as in Assumption 1. If $\|\mathbf{Y}_t\|^2 \le \frac{1}{\eta}$, then*

$$\|\nabla_x f(\mathbf{X}_t, \mathbf{Y}_t)\|_F^2 \le \frac{2}{\eta}\big(f(\mathbf{X}_t, \mathbf{Y}_t) - f(\mathbf{X}_{t+1}, \mathbf{Y}_t)\big). \tag{3}$$

*If moreover $\|\mathbf{X}_t\|^2 \le \frac{2}{\eta}$, then $f(\mathbf{X}_t, \mathbf{Y}_t) \le f(\mathbf{X}_t, \mathbf{Y}_{t-1})$. Consequently, if $\|\mathbf{Y}_t\|^2 \le \frac{1}{\eta}$ for all $t = 0, \ldots, T$, and $\|\mathbf{X}_t\|^2 \le \frac{2}{\eta}$ for all $t = 0, \ldots, T$, then $\sum_{t=0}^{T} \|\nabla_x f(\mathbf{X}_t, \mathbf{Y}_t)\|_F^2 \le \frac{2}{\eta} f(\mathbf{X}_0, \mathbf{Y}_0)$ Likewise, if $\|\mathbf{X}_{t+1}\|^2 \le \frac{1}{\eta}$, then*

$$\|\nabla_Y f(\mathbf{X}_{t+1}, \mathbf{Y}_t)\|_F^2 \le \frac{2}{\eta}(f(\mathbf{X}_{t+1}, \mathbf{Y}_t) - f(\mathbf{X}_{t+1}, \mathbf{Y}_{t+1})). \tag{4}$$

*and if $\|\mathbf{Y}_t\|^2 \le \frac{2}{\eta}$, then $f(\mathbf{X}_{t+1}, \mathbf{Y}_t) \le f(\mathbf{X}_t, \mathbf{Y}_t)$, and so if $\|\mathbf{X}_{t+1}\|^2 \le \frac{1}{\eta}$ for all $t = 0, \ldots, T$, and $\|\mathbf{Y}_t\|^2 \le \frac{2}{\eta}$ for all $t = 0, \ldots, T$, then $\sum_{t=0}^{T} \|\nabla_Y f(\mathbf{X}_{t+1}, \mathbf{Y}_t)\|_F^2 \le \frac{2}{\eta} f(\mathbf{X}_0, \mathbf{Y}_0)$.*

---

*Proof.* The proof of Lemma 4.1 is a direct calculation:

$$\begin{aligned}
f(\mathbf{X}_{t+1}, \mathbf{Y}_t) &= \frac{1}{2}\|\mathbf{A} - \mathbf{X}_{t+1}\mathbf{Y}_t^\intercal\|_F^2 \\
&= \frac{1}{2}\|\mathbf{A} - (\mathbf{X}_t - \eta\nabla_x f(\mathbf{X}_t, \mathbf{Y}_t))\mathbf{Y}_t^\intercal\|_F^2 \\
&= \frac{1}{2}\|\mathbf{A} - \mathbf{X}_t\mathbf{Y}_t^\intercal + \eta\nabla_x f(\mathbf{X}_t, \mathbf{Y}_t)\mathbf{Y}_t^\intercal\|_F^2 \\
&= \frac{1}{2}\|\mathbf{A} - \mathbf{X}_t\mathbf{Y}_t^\intercal\|_F^2 + \frac{1}{2}\|\eta\nabla_x f(\mathbf{X}_t, \mathbf{Y}_t)\mathbf{Y}_t^\intercal\|_F^2 - \eta\operatorname{trace}[(\mathbf{X}_t\mathbf{Y}_t^\intercal - \mathbf{A})(\nabla_x f(\mathbf{X}_t, \mathbf{Y}_t)\mathbf{Y}_t^\intercal)^\intercal] \\
&= \frac{1}{2}\|\mathbf{A} - \mathbf{X}_t\mathbf{Y}_t^\intercal\|_F^2 + \frac{1}{2}\|\eta\nabla_x f(\mathbf{X}_t, \mathbf{Y}_t)\mathbf{Y}_t^\intercal\|_F^2 - \eta\operatorname{trace}[\underbrace{(\mathbf{X}_t\mathbf{Y}_t^\intercal - \mathbf{A})\mathbf{Y}_t}_{\nabla_x f(\mathbf{X}_t, \mathbf{Y}_t)}(\nabla_x f(\mathbf{X}_t, \mathbf{Y}_t))^\intercal] \\
&= f(\mathbf{X}_t, \mathbf{Y}_t) + \frac{\eta}{2}\|\nabla_x f(\mathbf{X}_t, \mathbf{Y}_t)\mathbf{Y}_t^\intercal\|_F^2 - \eta\|\nabla_x f(\mathbf{X}_t, \mathbf{Y}_t)\|_F^2 \\
&\le f(\mathbf{X}_t, \mathbf{Y}_t) + \frac{\eta^2}{2}\|\nabla_x f(\mathbf{X}_t, \mathbf{Y}_t)\|_F^2\|\mathbf{Y}_t\|^2 - \eta\|\nabla_x f(\mathbf{X}_t, \mathbf{Y}_t)\|_F^2 \\
&\le f(\mathbf{X}_t, \mathbf{Y}_t) + \frac{\eta}{2}\|\nabla_x f(\mathbf{X}_t, \mathbf{Y}_t)\|_F^2 - \eta\|\nabla_x f(\mathbf{X}_t, \mathbf{Y}_t)\|_F^2 \\
&\le f(\mathbf{X}_t, \mathbf{Y}_t) - \frac{\eta}{2}\|\nabla_x f(\mathbf{X}_t, \mathbf{Y}_t)\|_F^2. \qquad \square
\end{aligned}$$

**Proposition 4.2** (Bounding singular values of iterates). *Consider alternating gradient descent as in [Assumption 1](). Set $f_0 := f(\mathbf{X}_0, \mathbf{Y}_0)$. Set $T_* = \left\lfloor \frac{1}{32\eta^2 f_0} \right\rfloor$. Suppose $\sigma_1^2(\mathbf{X}_0) \leq \frac{9}{16\eta}, \sigma_1^2(\mathbf{Y}_0) \leq \frac{9}{16\eta}$. Then for all $0 \leq T \leq T_*$,*

    *1. $\|\mathbf{X}_T\| \leq \frac{1}{\sqrt{\eta}}$ and $\|\mathbf{Y}_T\| \leq \frac{1}{\sqrt{\eta}}$,*

    *2. $\sigma_r(\mathbf{X}_0) - \sqrt{2T\eta f_0} \leq \sigma_r(\mathbf{X}_T) \leq \sigma_1(\mathbf{X}_T) \leq \sigma_1(\mathbf{X}_0) + \sqrt{2T\eta f_0}$,*

    *3. $\sigma_r(\mathbf{Y}_0) - \sqrt{2T\eta f_0} \leq \sigma_r(\mathbf{Y}_T) \leq \sigma_1(\mathbf{Y}_T) \leq \sigma_1(\mathbf{Y}_0) + \sqrt{2T\eta f_0}$.*

The proof of [Proposition 4.2]() is in the supplement [Appendix B]().

**Proposition 4.3** (Initialization). *Assume $\mathbf{X}_0$ and $\mathbf{Y}_0$ are initialized as in [Assumption 2](), which fixes $C \geq 1$, $\nu < 1$, and $D \leq \frac{C}{9}\nu$, and consider alternating gradient descent as in [Assumption 1](). Then with probability at least $1 - \delta$, with respect to the random initialization and $\delta$ defined in [(A2d)](),*

    *1. $\frac{1}{\sqrt{\eta}} \frac{\rho}{C} \frac{\sigma_r(\mathbf{A})}{\sigma_1(\mathbf{A})} \leq \sigma_r(\mathbf{X}_0), \quad \sigma_1(\mathbf{X}_0) \leq \frac{3}{C\sqrt{\eta}}, \quad \sigma_1(\mathbf{Y}_0) \leq \frac{\sqrt{\eta}\,C\,\nu\,\sigma_1(\mathbf{A})}{3}$,*

    *2. $\frac{1}{2(1-\nu)^2}\|\mathbf{A}\|_F^2 \leq f(\mathbf{X}_0, \mathbf{Y}_0) \leq \frac{1}{2}(1+\nu)^2\|\mathbf{A}\|_F^2$.*

The proof of [Proposition 4.3]() is in the supplement [Appendix C]().

Combining the previous two propositions gives the following.

**Corollary 4.4.** *Assume $\mathbf{X}_0$ and $\mathbf{Y}_0$ are initialized as in [Assumption 2](), with the stronger assumption that $C \geq 4$. Consider alternating gradient descent as in [Assumption 1]() with $\eta \leq \frac{9}{4C\nu\sigma_1(\mathbf{A})}$. With $\beta$ as in [(A2c)]() and $f_0 = f(\mathbf{X}_0, \mathbf{Y}_0)$, set $T = \left\lfloor \frac{\beta}{8\eta^2 f_0} \right\rfloor$. With probability at least $1 - \delta$, with respect to the random initialization and $\delta$ defined in [(A2d)](), the following hold for all $t = 1, \ldots, T$:*

$$\sigma_r(\mathbf{X}_t) \geq \frac{1}{2}\sqrt{\frac{\beta}{\eta}}, \text{ and } \sigma_1(\mathbf{X}_t), \sigma_1(\mathbf{Y}_t) \leq \frac{3}{C\sqrt{\eta}} + \frac{1}{2}\sqrt{\frac{\beta}{\eta}}$$

*Proof.* By [Proposition 4.3](), we have the following event occurring with the stated probability:

$$\frac{\rho^2\sigma_r^2(\mathbf{A})}{C^2\sigma_1^2(\mathbf{A})\eta} \leq \sigma_r^2(\mathbf{X}_0) \leq \sigma_1^2(\mathbf{X}_0) \leq \frac{9}{16\eta}$$

where the upper bound uses that $C \geq 4$. Moreover, using that $\eta \leq \frac{9}{4C\nu\sigma_1(\mathbf{A})}$, $\sigma_1^2(\mathbf{Y}_0) \leq 9\eta C^2\sigma_1(\mathbf{A})^2 \leq \frac{9}{16\eta}$. For $\beta$ as in [(A2c)](), note that $T = \left\lfloor \frac{\beta}{8\eta^2 f_0} \right\rfloor \leq \left\lfloor \frac{1}{32\eta^2 f_0} \right\rfloor$, which means that we can apply [Proposition 4.2]() up to iteration $T$, resulting in the bound $\sigma_r(\mathbf{X}_t) \geq \sigma_r(\mathbf{X}_0) - \sqrt{2T\eta f_0} \geq \frac{1}{2}\sqrt{\frac{\beta}{\eta}}$. Similarly, $\sigma_1(\mathbf{X}_t), \sigma_1(\mathbf{Y}_t) \leq \frac{3}{C\sqrt{\eta}} + \frac{1}{2}\sqrt{\frac{\beta}{\eta}}$. $\qquad\square$

Finally, we use a couple crucial lemmas which apply to our initialization of $\mathbf{X}_0$ and $\mathbf{Y}_0$.

**Lemma 4.5.** *Consider alternating gradient descent as in [Assumption 1](). If $ColSpan(\mathbf{X}_0) \subseteq ColSpan(\mathbf{A})$, then $ColSpan(\mathbf{X}_t) \subseteq ColSpan(\mathbf{A})$ for all $t$.*

*Proof.* Suppose $\mathrm{ColSpan}(\mathbf{X}_t) \subseteq \mathrm{ColSpan}(\mathbf{A})$. Then $\mathrm{ColSpan}(\mathbf{X}_t\mathbf{Y}_t^\mathsf{T}\mathbf{Y}_t) \subseteq \mathrm{ColSpan}(\mathbf{X}_t) \subseteq \mathrm{ColSpan}(\mathbf{A})$ and by the update of [Assumption 1](),

$$\begin{aligned}
\mathrm{ColSpan}(\mathbf{X}_{t+1}) &= \mathrm{ColSpan}(\mathbf{X}_t + \eta\mathbf{A}\mathbf{Y}_t - \eta\mathbf{X}_t\mathbf{Y}_t^\mathsf{T}\mathbf{Y}_t) \\
&\subseteq \mathrm{ColSpan}(\mathbf{X}_t) \cup \mathrm{ColSpan}(\mathbf{A}\mathbf{Y}_t) \cup \mathrm{ColSpan}(\mathbf{X}_t\mathbf{Y}_t^\mathsf{T}\mathbf{Y}_t) \\
&\subseteq \mathrm{ColSpan}(\mathbf{A}). \qquad\qquad\qquad\qquad\qquad\qquad\qquad\square
\end{aligned}$$

> **Lemma 4.6.** *If $\mathbf{A}$ is rank $r$, $ColSpan(\mathbf{X}_t) \subseteq ColSpan(\mathbf{A})$, and $\sigma_r(\mathbf{X}_t) > 0$ then*
>
> $$\|\nabla_{\mathbf{Y}} f(\mathbf{X}_t, \mathbf{Y}_{t-1})\|_{\mathrm{F}}^2 \geq 2\sigma_r^2(\mathbf{X}_t) f(\mathbf{X}_t, \mathbf{Y}_{t-1}). \tag{5}$$

*Proof.* If $\mathbf{A}$ is rank $r$, $\mathrm{ColSpan}(\mathbf{X}_t) \subseteq \mathrm{ColSpan}(\mathbf{A})$, and $\sigma_r(\mathbf{X}_t) > 0$, then $\mathbf{X}_t$ is rank-$r$; thus, $\mathrm{ColSpan}(\mathbf{X}_t) = \mathrm{ColSpan}(\mathbf{A})$. In this case, each column of $(\mathbf{X}_t \mathbf{Y}_t^\intercal - \mathbf{A})$ is in the row span of $\mathbf{X}_t^\intercal$, and so

$$\|\nabla_{\mathbf{Y}} f(\mathbf{X}_t, \mathbf{Y}_{t-1})\|_{\mathrm{F}}^2 = \|(\mathbf{X}_t \mathbf{Y}_{t-1}^\intercal - \mathbf{A})^\intercal \mathbf{X}_t\|_{\mathrm{F}}^2$$
$$= \|\mathbf{X}_t^\intercal (\mathbf{X}_t \mathbf{Y}_{t-1}^\intercal - \mathbf{A})\|_{\mathrm{F}}^2 \geq \sigma_r^2(\mathbf{X}_t) \|\mathbf{X}_t \mathbf{Y}_{t-1}^\intercal - \mathbf{A}\|_{\mathrm{F}}^2. \quad \square$$

*Remark* 4.7. While Lemmas 4.5 and 4.6 are straightforward to prove in the setting we consider where $\mathbf{A}$ is exactly rank-$r$, these lemmas no longer hold beyond the exactly rank-$r$ setting (while the rest of the theorems we use do extend). Numerical experiments such as Figure 4 illustrate that the proposed algorithm does extend to finding best low-rank approximations to general matrices, but the theory for these experiments will require a careful reworking of Lemmas 4.5 and 4.6.

# 5 Main results

We are now ready to prove the main results.

> **Theorem 5.1.** *Assume $\mathbf{X}_0$ and $\mathbf{Y}_0$ are initialized as in Assumption 2, with the stronger assumption that $C \geq 4$. Consider alternating gradient descent as in Assumption 1 with*
>
> $$\eta \leq \frac{9}{4C\nu\sigma_1(\mathbf{A})}.$$
>
> *With $\beta$ as in (A2c) and $f_0 = f(\mathbf{X}_0, \mathbf{Y}_0)$, set*
>
> $$T = \left\lfloor \frac{\beta}{8\eta^2 f_0} \right\rfloor.$$
>
> *Then with probability at least $1 - \delta$, with respect to the random initialization and $\delta$ defined in (A2d), the following hold for all $t = 1, \ldots, T$:*
>
> $$\|\mathbf{A} - \mathbf{X}_t \mathbf{Y}_t^\intercal\|_{\mathrm{F}}^2 \leq 2 \exp\left(-\beta t/4\right) f_0$$
> $$\leq \exp\left(-\beta t/4\right) (1 + \nu)^2 \|\mathbf{A}\|_{\mathrm{F}}^2. \tag{6}$$

*Proof.* Corollary 4.4 implies that $\sigma_r(\mathbf{X}_t)^2 \geq \frac{\beta}{4\eta}$ for $t = 1, \ldots, T$. Lemmas 4.5 and 4.6 imply since $\mathbf{X}_0$ is initialized in the column space of $\mathbf{A}$, $\mathbf{X}_t$ remains in the column space of $\mathbf{A}$ for all $t$, and

$$\|\nabla_{\mathbf{Y}} f(\mathbf{X}_{t+1}, \mathbf{Y}_t)\|_{\mathrm{F}}^2 = \|(\mathbf{A}^\intercal - \mathbf{Y}_t \mathbf{X}_{t+1}^\intercal)\mathbf{X}_{t+1}\|_{\mathrm{F}}^2 \geq \sigma_r(\mathbf{X}_{t+1})^2 \|(\mathbf{A}^\intercal - \mathbf{Y}_t \mathbf{X}_{t+1}^\intercal)\|_{\mathrm{F}}^2$$
$$\geq \frac{\beta}{4\eta}\|\mathbf{A} - \mathbf{X}_{t+1} \mathbf{Y}_t^\intercal\|_{\mathrm{F}}^2 = \frac{\beta}{2\eta} f(\mathbf{X}_{t+1}, \mathbf{Y}_t). \tag{7}$$

That is, a lower bound on $\sigma_r(\mathbf{X}_t)^2$ implies that the gradient step with respect to $\mathbf{Y}$ satisfies the Polyak-Lojasiewicz (PL)-equality[3].

We can combine this PL inequality with the Lipschitz bound from Lemma 4.1 to derive the linear convergence rate. Indeed, by (3),

$$f(\mathbf{X}_{t+1}, \mathbf{Y}_{t+1}) - f(\mathbf{X}_{t+1}, \mathbf{Y}_t) \leq -\frac{\eta}{2}\|\nabla_{\mathbf{Y}} f(\mathbf{X}_{t+1}, \mathbf{Y}_t)\|_{\mathrm{F}}^2 \leq -\frac{\beta}{4} f(\mathbf{X}_{t+1}, \mathbf{Y}_t).$$

---

[3]A function $f$ satisfies the PL-equality if for all $\mathbf{x} \in \mathbb{R}^m$, $f(\mathbf{x}) - f(\mathbf{x}^*) \leq \frac{1}{2m}\|\nabla f(\mathbf{x})\|^2$, where $f(\mathbf{x}^*) = \min_{\mathbf{x}} f(\mathbf{x})$

where the final inequality is (7). Consequently, using Proposition 4.3,

$$f(\mathbf{X}_T, \mathbf{Y}_T) \le (1 - \beta/4)f(\mathbf{X}_{T-1}, \mathbf{Y}_{T-1}) \le (1 - \beta/4)^T f(\mathbf{X}_0, \mathbf{Y}_0)$$
$$\le \exp\left(-\beta T/4\right) f(\mathbf{X}_0, \mathbf{Y}_0). \quad \square$$

---

**Corollary 5.2.** *Assume $\mathbf{X}_0$ and $\mathbf{Y}_0$ are initialized as in Assumption 2, with the stronger assumptions that $C \ge 4$ and $\nu \le \frac{1}{2}$. Consider alternating gradient descent as in Assumption 1 with*

$$\eta \le \frac{\beta}{\sqrt{32 f_0 \log(2 f_0/\epsilon)}}, \tag{8}$$

*where $\beta$ is defined in (A2c) and $f_0 = f(\mathbf{X}_0, \mathbf{Y}_0)$. Then with probability at least $1 - \delta$, with respect to the random initialization and $\delta$ defined in (A2d), it holds*

$$\|\mathbf{A} - \mathbf{X}_T \mathbf{Y}_T^\mathsf{T}\|_\mathrm{F}^2 \le \epsilon \quad \text{at interation} \quad T = \left\lfloor \frac{\beta}{8\eta^2 f_0} \right\rfloor.$$

*Here $\rho$ is defined in (A2b). Using the upper bound for $\eta$ in (8), the iteration complexity to reach an $\epsilon$-optimal loss value is*

$$T = \mathcal{O}\left( \left( \frac{\sigma_1(\mathbf{A})}{\sigma_r(\mathbf{A})} \right)^2 \frac{1}{\rho^2} \log\left( \frac{\|\mathbf{A}\|_\mathrm{F}^2}{\epsilon} \right) \right).$$

---

This corollary follows from Theorem 5.1 by solving for $\eta$ so that the RHS of (6) is at most $\epsilon$, and then noting that $\eta \le \frac{\beta}{\sqrt{32 f_0 \log(2 f_0/\epsilon)}}$ implies that $\eta \le \frac{9}{4C\nu\sigma_1(\mathbf{A})}$ when $\nu \le \frac{1}{2}$, using the lower bound on $f_0$ from Proposition 4.3.

Using this corollary recursively, we can prove that the loss value remains small for $T' \ge \lfloor \frac{\beta}{8\eta^2 f_0} \rfloor$, provided we increase the lower bound on $C$ by a factor of 2. The proof is in supplementary section D.

---

**Corollary 5.3.** *Assume $\mathbf{X}_0$ and $\mathbf{Y}_0$ are initialized as in Assumption 2, with the stronger assumptions that $C \ge 8$ and $\nu \le \frac{1}{2}$. Fix $\epsilon < 1/16$, and consider alternating gradient descent as in Assumption 1 with*

$$\eta \le \frac{\beta}{\sqrt{32 f_0 \log(1/\epsilon)}}, \tag{9}$$

*where $\beta$ is defined in (A2c) and $f_0 = f(\mathbf{X}_0, \mathbf{Y}_0)$. Then with probability at least $1 - \delta$, with respect to the random initialization and $\delta$ defined in (A2d), it holds for any $k \in \mathbb{N}$ that*

$$\|\mathbf{A} - \mathbf{X}_T \mathbf{Y}_T^\mathsf{T}\|_\mathrm{F}^2 \le \epsilon^k \|\mathbf{A} - \mathbf{X}_0 \mathbf{Y}_0^\mathsf{T}\|_\mathrm{F}^2 \qquad \text{for} \quad T \ge \sum_{\ell=0}^{k-1} \left\lfloor \left( \frac{1}{4\epsilon} \right)^\ell \frac{\beta}{8\eta^2 f_0} \right\rfloor.$$

---

# 6 Numerical experiments

We perform an illustrative numerical experiment to demonstrate both the theoretical and practical benefits of the proposed initialization. We use gradient descent *without* alternating to demonstrate that this theoretical assumption makes little difference in practice. We factorize a rank-5 ($r = 5$) matrix of size $100 \times 100$. The matrix is constructed as $\mathbf{A} = \mathbf{U}\boldsymbol{\Sigma}\mathbf{V}^\mathsf{T}$ with $\mathbf{U}$ and $\mathbf{V}$ random $100 \times 5$ orthonormal matrices and singular value ratio $\sigma_r(\mathbf{A})/\sigma_1(\mathbf{A}) = 0.9$. The same matrix is used for

each set of experiments. We compare four initializations:

$$\text{Proposed:} \qquad \mathbf{X}_0 = \frac{1}{\sqrt{\eta}\sqrt{d}C\sigma_1}\mathbf{A}\boldsymbol{\Phi}_{(n\times d)} \qquad \mathbf{Y}_0 = \frac{\sqrt{\eta}D\sigma_1}{\sqrt{n}}\boldsymbol{\Phi}_{(n\times d)}$$

$$\text{ColSpan}(\mathbf{A})\text{:} \qquad \mathbf{X}_0 = \frac{1}{10\sqrt{d}}\mathbf{A}\boldsymbol{\Phi}_{(n\times d)} \qquad \mathbf{Y}_0 = \frac{1}{10\sqrt{n}}\boldsymbol{\Phi}_{(n\times d)}$$

$$\text{Random:} \qquad \mathbf{X}_0 = \frac{1}{10\sqrt{m}}\boldsymbol{\Phi}_{(m\times d)} \qquad \mathbf{Y}_0 = \frac{1}{10\sqrt{n}}\boldsymbol{\Phi}_{(n\times d)}$$

$$\text{Random-Asym:} \qquad \mathbf{X}_0 = \frac{1}{\sqrt{\eta}}\frac{1}{10\sqrt{m}}\boldsymbol{\Phi}_{(m\times d)} \qquad \mathbf{Y}_0 = \sqrt{\eta}\frac{1}{10\sqrt{n}}\boldsymbol{\Phi}_{(n\times d)}$$

Here, $\boldsymbol{\Phi}$ denotes a random matrix with independent entries from $\mathcal{N}(0,1)$. The random initialization is what is commonly used and analyzed. We include ColSpan($\mathbf{A}$) to understand the impact of starting in the column space of $\mathbf{A}$ for $\mathbf{X}_0$. Our proposed initialization (Assumption 2) combines this with an asymmetric scaling. In all experiments we use the defaults $C = 4$ and $D = C\nu/9$ with $\nu = 1e{-}10$ for computing the proposed initialization as well as the theoretical step size. We assume $\sigma_1 = \sigma_1(\mathbf{A})$ is known in these cases. (In practice, a misestimate of $\sigma_1$ can be compensated with a different value for $C$.)

Figure 2 shows how the proposed method performs using the theoretical step size and compared to its theory. We also compare our initialization to three other initializations with the same step size. We consider two different levels of over-factoring, choosing $d = 10$ (i.e, $2r$) and $d = 6$ (i.e., $r + 1$). All methods perform better for larger $d = 10$. The theory for the proposed initialization underestimates its performance (there may be room for further improvement) but still shows a stark and consistent advantage compared to the performance of the standard initialization, as well as compared to initializing in the column span of $\mathbf{A}$ but not asymmetrically, or initializing asymmetrically but not in the column span of $\mathbf{A}$.

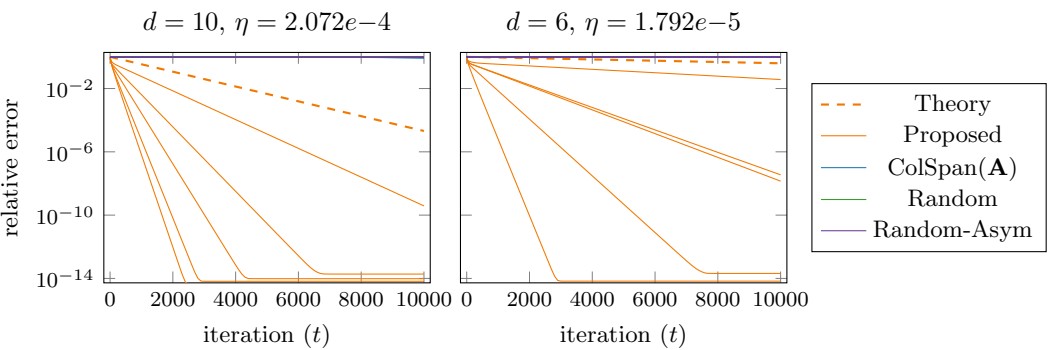

Figure 2: **Theoretical advantage of proposed initialization.** Gradient descent (non-alternating) for $\mathbf{A} \in \mathbb{R}^{100\times 100}$ with rank($\mathbf{A}$) = 5, $\sigma_1 = 1$ and $\sigma_r = 0.9$. The plot shows five runs with each type of initialization.

Figure 3 shows how the three initializations compare with different step lengths. The advantage of the proposed initialization persists even with step lengths that are larger than that proposed by the theory, up until the step length is too large for any method to converge ($\eta = 1$). We emphasize that we are showing standard gradient descent, not alternating gradient descent. (There is no major difference between the two in our experience.)

Although the theory requires that $\mathbf{A}$ be exactly rank-$r$, Fig. 4 shows that the proposed initialization still maintains its advantage for noisy problems that are only approximately rank-$r$.

## References

[ABC$^+$22] Kwangjun Ahn, Sébastien Bubeck, Sinho Chewi, Yin Tat Lee, Felipe Suarez, and Yi Zhang. Learning threshold neurons via the "edge of stability". *arXiv preprint arXiv:2212.07469*, December 2022.

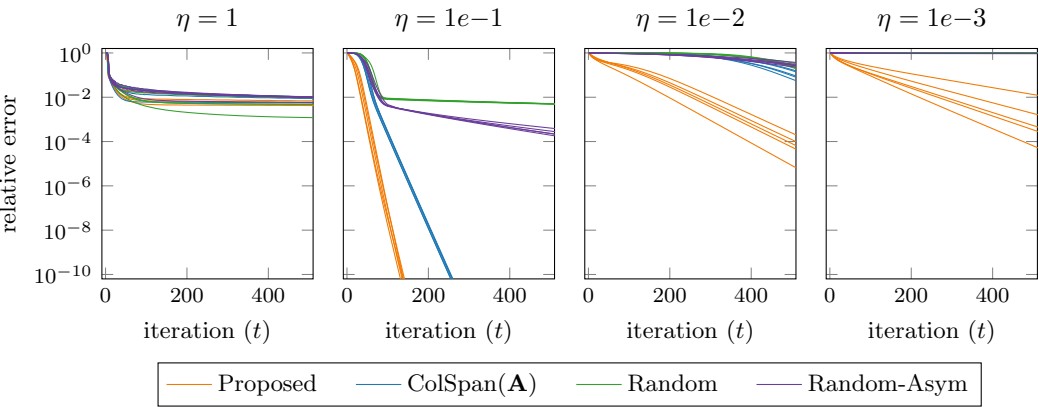

Figure 3: **Advantage of proposed initialization for different step lengths.** Gradient descent (non-alternating) for $\mathbf{A} \in \mathbb{R}^{100 \times 100}$ with rank$(\mathbf{A}) = 5$, $\sigma_1 = 1$ and $\sigma_r = 0.9$. The plot shows five runs with each type of initialization.

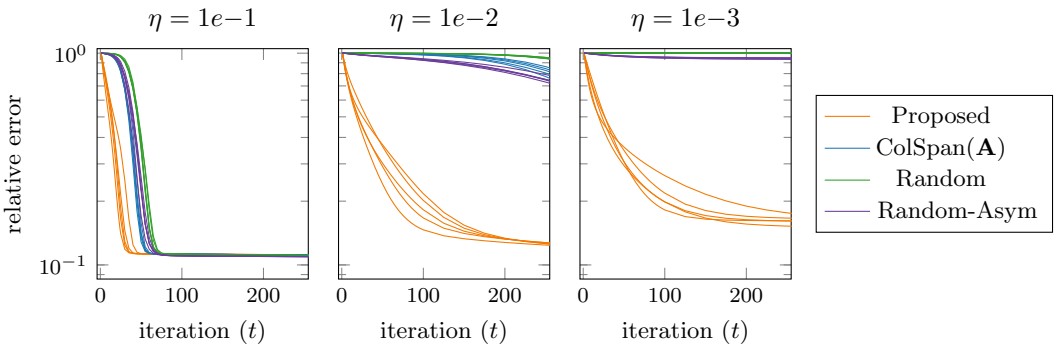

Figure 4: **Advantage of proposed initialization for noisy problems.** Gradient descent (non-alternating) for $\mathbf{A} \in \mathbb{R}^{100 \times 100}$ with rank$(\mathbf{A}) \approx 5$ (10% noise), $\sigma_1 = 1$ and $\sigma_r = 0.9$. The plot shows five runs with each type of initialization.

[ACHL19] Sanjeev Arora, Nadav Cohen, Wei Hu, and Yuping Luo. Implicit regularization in deep matrix factorization. *Advances in Neural Information Processing Systems*, 32, 2019.

[BKS16] Srinadh Bhojanapalli, Anastasios Kyrillidis, and Sujay Sanghavi. Dropping convexity for faster semi-definite optimization. In *Conference on Learning Theory*, pages 530–582. PMLR, 2016.

[BM03] Samuel Burer and Renato DC Monteiro. A nonlinear programming algorithm for solving semidefinite programs via low-rank factorization. *Mathematical Programming*, 95(2):329–357, 2003.

[BM05] Samuel Burer and Renato DC Monteiro. Local minima and convergence in low-rank semidefinite programming. *Mathematical programming*, 103(3):427–444, 2005.

[CB22] Lei Chen and Joan Bruna. On gradient descent convergence beyond the edge of stability. *arXiv preprint arXiv:2206.04172*, June 2022.

[CCFM19] Yuxin Chen, Yuejie Chi, Jianqing Fan, and Cong Ma. Gradient descent with random initialization: Fast global convergence for nonconvex phase retrieval. *Mathematical Programming*, 176:5–37, 2019.

[CGMR20] Hung-Hsu Chou, Carsten Gieshoff, Johannes Maly, and Holger Rauhut. Gradient descent for deep matrix factorization: Dynamics and implicit bias towards low rank. *arXiv preprint arXiv:2011.13772*, 2020.

[CKL+21] Jeremy M Cohen, Simran Kaur, Yuanzhi Li, J Zico Kolter, and Ameet Talwalkar. Gradient descent on neural networks typically occurs at the edge of stability. *arXiv preprint arXiv:2103.00065*, February 2021.

[DHL18] Simon S Du, Wei Hu, and Jason D Lee. Algorithmic regularization in learning deep homogeneous models: Layers are automatically balanced. *Advances in neural information processing systems*, 31, 2018.

[GHJY15] Rong Ge, Furong Huang, Chi Jin, and Yang Yuan. Escaping from saddle points—online stochastic gradient for tensor decomposition. In *Conference on learning theory*, pages 797–842. PMLR, 2015.

[GWB+17] Suriya Gunasekar, Blake E Woodworth, Srinadh Bhojanapalli, Behnam Neyshabur, and Nati Srebro. Implicit regularization in matrix factorization. *Advances in Neural Information Processing Systems*, 30, 2017.

[HKD20] David Hong, Tamara G Kolda, and Jed A Duersch. Generalized canonical polyadic tensor decomposition. *SIAM Review*, 62(1):133–163, 2020.

[JCD22] Liwei Jiang, Yudong Chen, and Lijun Ding. Algorithmic regularization in model-free overparametrized asymmetric matrix factorization. *arXiv preprint arXiv:2203.02839*, March 2022.

[JJKN17] Prateek Jain, Chi Jin, Sham Kakade, and Praneeth Netrapalli. Global convergence of non-convex gradient descent for computing matrix squareroot. In *Artificial Intelligence and Statistics*, pages 479–488. PMLR, 2017.

[KH20] Tamara G Kolda and David Hong. Stochastic gradients for large-scale tensor decomposition. *SIAM Journal on Mathematics of Data Science*, 2(4):1066–1095, 2020.

[LMZ18] Yuanzhi Li, Tengyu Ma, and Hongyang Zhang. Algorithmic regularization in over-parameterized matrix sensing and neural networks with quadratic activations. In *Conference On Learning Theory*, pages 2–47. PMLR, 2018.

[RV09] Mark Rudelson and Roman Vershynin. Smallest singular value of a random rectangular matrix. *Communications on Pure and Applied Mathematics: A Journal Issued by the Courant Institute of Mathematical Sciences*, 62(12):1707–1739, 2009.

[SWW17] Sujay Sanghavi, Rachel Ward, and Chris D White. The local convexity of solving systems of quadratic equations. *Results in Mathematics*, 71:569–608, 2017.

[TBS+16] Stephen Tu, Ross Boczar, Max Simchowitz, Mahdi Soltanolkotabi, and Ben Recht. Low-rank solutions of linear matrix equations via Procrustes flow. In *International Conference on Machine Learning*, pages 964–973. PMLR, 2016.

[Ver10] Roman Vershynin. Introduction to the non-asymptotic analysis of random matrices. In *Compressed Sensing*, pages 210–268. Cambridge University Press, may 2010.

[WCZT21] Yuqing Wang, Minshuo Chen, Tuo Zhao, and Molei Tao. Large learning rate tames homogeneity: Convergence and balancing effect. *arXiv preprint arXiv:2110.03677*, 2021.

[YD21] Tian Ye and Simon S Du. Global convergence of gradient descent for asymmetric low-rank matrix factorization. *Advances in Neural Information Processing Systems*, 34:1429–1439, 2021.

[ZL16] Qinqing Zheng and John Lafferty. Convergence analysis for rectangular matrix completion using Burer-Monteiro factorization and gradient descent. *arXiv preprint arXiv:1605.07051*, 2016.

[ZWL15] Tuo Zhao, Zhaoran Wang, and Han Liu. Nonconvex low rank matrix factorization via inexact first order oracle. *Advances in Neural Information Processing Systems*, 458:461–462, 2015.

## A   Non-asymptotic singular value bounds

**Proposition A.1** (Theorem 1.1 of [RV09]). *Let $\mathbf{A}$ be an $d \times r$ matrix, $d \geq r$, whose entries are independently drawn from $\mathcal{N}(0,1)$. Then for every $\tau \geq 0$,*

$$Pr\left(\sigma_r(\mathbf{A}) \leq \tau(\sqrt{d} - \sqrt{r-1})\right) \leq (C_1\tau)^{d-r+1} + e^{-C_2 d}$$

*where $C_1, C_2 > 0$ are universal constants.*

**Proposition A.2** ([Ver10]). *Let $\mathbf{A}$ be an $d \times r$ matrix whose entries are independently drawn from $\mathcal{N}(0,1)$. Then for every $t \geq 0$, with probability at least $1 - \exp(-t^2/2)$, we have*

$$\sigma_r(\mathbf{A}) \geq \sqrt{d} - \sqrt{r} - t$$

*and for every $t \geq 0$, with probability at least $1 - \exp(-t^2/2)$, we have*

$$\sigma_1(\mathbf{A}) \leq \sqrt{d} + \sqrt{r} + t$$

## B   Proof of Proposition 4.2

First, observe that by assumption, $\|\mathbf{X}_0\|^2, \|\mathbf{Y}_0\|^2 \leq \frac{9}{16\eta} \leq \frac{1}{\eta}$. Now, suppose that that $\|\mathbf{X}_0\|^2, \|\mathbf{Y}_0\|^2 \leq \frac{9}{16\eta}$ and $\|\mathbf{X}_t\|^2, \|\mathbf{Y}_t\|^2 \leq \frac{1}{\eta}$ for $t = 0, \ldots T-1$, and $1 \leq T \leq \lfloor\frac{1}{32\eta^2 f_0}\rfloor$. Then by Lemma 4.1,

$$\sum_{t=0}^{T-1}\left\|\nabla_{\mathrm{x}}f(\mathbf{X}_t, \mathbf{Y}_t)\right\|_{\mathrm{F}}^2 \leq \frac{2}{\eta}f_0. \tag{10}$$

Hence,

$$\|\mathbf{X}_T - \mathbf{X}_0\| \leq \eta\left\|\sum_{t=0}^{T-1}\nabla_{\mathrm{x}}f(\mathbf{X}_t, \mathbf{Y}_t)\right\| \leq \eta\left\|\sum_{t=0}^{T-1}\nabla_{\mathrm{x}}f(\mathbf{X}_t, \mathbf{Y}_t)\right\|_{\mathrm{F}}$$

$$\leq \eta\sqrt{\sum_{t=0}^{T-1}\|\nabla_{\mathrm{x}}f(\mathbf{X}_t, \mathbf{Y}_t)\|_F^2} \leq \eta\sqrt{\frac{2}{\eta}f_0} = \sqrt{2\eta f_0}. \tag{11}$$

Then, for $T \leq T_*$, $\|\mathbf{X}_T\| \leq \|\mathbf{X}_0\| + \|\mathbf{X}_T - \mathbf{X}_0\| \leq \frac{3}{4\sqrt{\eta}} + \sqrt{2T\eta f_0} \leq \frac{1}{\sqrt{\eta}}$. It follows that $\|\mathbf{X}_t\|^2 \leq \frac{1}{\eta}$ for $t = 0, \ldots T$. Using Lemma 4.1 again, repeating the same argument,

$$\|\mathbf{Y}_t\| \leq \frac{1}{\sqrt{\eta}}, \quad t = 0, \ldots, T.$$

Iterate the induction until $T = T_* = \lfloor\frac{1}{32\eta^2 f_0}\rfloor$, to obtain $\|\mathbf{X}_t\|^2, \|\mathbf{Y}_t\|^2 \leq \frac{1}{\eta}$ for $t = 1, \ldots, T_*$.

Because $\|\mathbf{X}_T - \mathbf{X}_0\| \leq \sqrt{2\eta T f_0}$ for $T \leq T_* = \lfloor\frac{1}{32\eta^2 f_0}\rfloor$,

$$\sigma_r(\mathbf{X}_T) \geq \sigma_r(\mathbf{X}_0) - \|\mathbf{X}_T - \mathbf{X}_0\|; \qquad \sigma_1(\mathbf{X}_T) \leq \sigma_1(\mathbf{X}_0) + \|\mathbf{X}_T - \mathbf{X}_0\|.$$

A similar argument applies to achieve the stated bounds for $\sigma_r(\mathbf{Y}_T)$ and $\sigma_1(\mathbf{Y}_T)$.

## C   Proof of Proposition 4.3

Write the SVD $\mathbf{A} = \mathbf{U}_{m \times r}\mathbf{\Sigma}_{r \times r}\mathbf{V}_{r \times n}^{\mathsf{T}}$ so that $\mathbf{A}\mathbf{\Phi}_1 = \mathbf{U}_{m \times r}\mathbf{\Sigma}_{r \times r}(\mathbf{V}^{\mathsf{T}}\mathbf{\Phi}_1)$. Note that $\mathbf{V}^{\mathsf{T}}\mathbf{\Phi}_1 \in \mathbb{R}^{r \times d}$ has i.i.d. Gaussian entries $\mathcal{N}(0, \frac{1}{d})$. By Proposition A.1, with probability at least $1 - (C_1\epsilon)^{d-r+1} - e^{-C_2 d}$,

$$\sigma_r(\mathbf{V}^{\mathsf{T}}\mathbf{\Phi}_1) \geq \epsilon\left(1 - \frac{\sqrt{r-1}}{\sqrt{d}}\right)$$

On the other hand, Proposition A.2 implies that with probability at least $1 - e^{-r/2} - e^{-d/2}$,

$$\sigma_1(\mathbf{\Phi}_1) \leq \left(1 + \frac{2\sqrt{r}}{\sqrt{d}}\right) \leq 3, \quad \text{and} \quad \sigma_1(\mathbf{\Phi}_2) \leq \left(1 + \frac{2\sqrt{d}}{\sqrt{m}}\right) \leq 3.$$

If all aforementioned events hold, $\sigma_1(\mathbf{V}^\intercal \mathbf{\Phi}_1) \leq \sigma_1(\mathbf{V})\sigma_1(\mathbf{\Phi}_1) \leq 3$, and

$$\frac{\epsilon\left(1 - \frac{\sqrt{r-1}}{\sqrt{d}}\right)}{\sqrt{\eta}C\sigma_1(\mathbf{A})}\sigma_r(\mathbf{A}) \leq \sigma_r(\mathbf{X}_0) \leq \sigma_1(\mathbf{X}_0) \leq \frac{3}{\sqrt{\eta}C}, \quad \sigma_1(\mathbf{Y}_0) \leq 3\sqrt{\eta}D\sigma_1(\mathbf{A}) \leq \frac{\sqrt{\eta}C\nu\sigma_1(\mathbf{A})}{3}.$$

where the last inequality uses $D \leq \frac{C\nu}{9}$. Consequently,

$$1 - \nu \leq 1 - \frac{D}{C}\sigma_1(\mathbf{\Phi}_1)\sigma_1(\mathbf{\Phi}_2) \leq \left\|\mathbf{I} - \frac{D}{C}\mathbf{\Phi}_1\mathbf{\Phi}_2^\intercal\right\| \leq 1 + \frac{D}{C}\sigma_1(\mathbf{\Phi}_1)\sigma_1(\mathbf{\Phi}_2) \leq 1 + \nu.$$

Hence,

$$2f(\mathbf{X}_0, \mathbf{Y}_0) = \left\|\mathbf{A}(\mathbf{I} - \frac{D}{C}\mathbf{\Phi}_1\mathbf{\Phi}_2^\intercal)\right\|_{\mathrm{F}}^2 \leq (1+\nu)^2\|\mathbf{A}\|_{\mathrm{F}}^2$$

$$2f(\mathbf{X}_0, \mathbf{Y}_0) \geq \sigma_{\min}^2(\mathbf{I} - \frac{D}{C}\mathbf{\Phi}_1\mathbf{\Phi}_2^\intercal)\|\mathbf{A}\|_F^2 \geq \frac{1}{(1-\nu)^2}\|\mathbf{A}\|_F^2 \quad (12)$$

## D   Proof of Corollary 5.3

Set $\beta_1 = \beta$ as in (A2c). Set $f_{0(1)} = f_0$.

By Corollary 5.2, iterating Assumption 1 for $T_1 = \lfloor\frac{\beta_1}{8\eta^2 f_{0(1)}}\rfloor$ iterations with step-size

$$\eta \leq \frac{\beta_1}{\sqrt{32f_{0(1)}\log(1/\epsilon)}}$$

guarantees that

$$\frac{1}{2}\|\mathbf{A} - \mathbf{X}_{T_1}\mathbf{Y}'_{T_1}\|_{\mathrm{F}}^2 \leq f_{0(2)} := \epsilon f_{0(1)};$$

$$\|\sigma_r(\mathbf{X}_{T_1})\|^2 \geq \frac{1}{4}\frac{\beta_1}{\eta}.$$

This means that at time $T_1$, we can restart the analysis, and appeal again to Proposition 4.2 with modified parameters

- $f(\mathbf{X}_{T_1}, \mathbf{Y}_{T_1}) \leq f_{02} := \epsilon f_{01}$,
- $\beta_2 := \frac{\beta_1}{4}$.

Corollary 5.2 again guarantees that provided

$$\eta \leq \frac{\beta_2}{\sqrt{32f_{0(2)}\log(1/\epsilon)}} = \frac{1}{4\sqrt{\epsilon}}\frac{\beta_1}{\sqrt{32f_{0(1)}\log(1/\epsilon)}} \quad (13)$$

then $f(\mathbf{X}_{T_1+T_2}, \mathbf{Y}_{T_1+T_2}) \leq \epsilon f(\mathbf{X}_{T_1}, \mathbf{Y}_{T_1}) \leq \epsilon^2 f(\mathbf{X}_0, \mathbf{Y}_0)$ where

$$T_2 = \frac{T_1}{4\epsilon}. \quad (14)$$

We have that (13) is satisfied by assumption as we assume $\epsilon \leq \frac{1}{16}$. Repeating this inductively, we find that after $T = T_1 + \cdots + T_k = T_1\sum_{\ell=0}^{k-1}(\frac{1}{4\epsilon})^\ell \leq T_1(\frac{1}{4\epsilon})^k$ iterations, we are guaranteed that $f(\mathbf{X}_T, \mathbf{Y}_T) \leq \epsilon^k f(\mathbf{X}_0, \mathbf{Y}_0)$. This is valid for any $T \in \mathbb{N}$ because we may always apply

Proposition 4.2 in light of summability and $C \geq 8$: for any $t$,

$$\sigma_1(\mathbf{X}_t) \leq \sigma_1(\mathbf{X}_0) + \sqrt{\eta} \sum_{j=1}^{k} \sqrt{2T_k f_{0(k)}}$$

$$\leq \frac{3}{8\sqrt{\eta}} + \frac{1}{\sqrt{\eta}} \sum_{j=1}^{k} \sqrt{2(1/(4\epsilon))^j \frac{\beta_1}{8f_{0_1}} \epsilon^j f_{0(1)}}$$

$$\leq \frac{3}{8\sqrt{\eta}} + \frac{\sqrt{\beta}}{2\sqrt{\eta}} \sum_{j=1}^{k} (1/2)^j$$

$$\leq \frac{3 + 4\sqrt{\beta}}{8\sqrt{\eta}} \leq \frac{1}{2\sqrt{\eta}}.$$

