# OpenReview forum: "Convergence of Alternating Gradient Descent for Matrix Factorization"
_NeurIPS.cc/2023/Conference — NeurIPS 2023 spotlight_

### Official Review · Reviewer_GCxr · 2023-06-16

**Soundness:** 3 good
**Presentation:** 3 good
**Contribution:** 1 poor
**Rating:** 4
**Confidence:** 4

**Summary:**

This problem considers a matrix factorization problem: it seeks to minimize a function of the form $f(X,Y) = 1/2 \Vert A - XY^T \Vert_F^2$. In general, matrix factorization problems have applications to matrix sensing, phase retrieval, and are seen as prototypical non-convex optimization problems. The special case of the matrix factorization problem provided in this paper is very simple, and should be seen as a first theoretical step to understand other matrix factorization problems with more practical applications.

The contribution of this paper is to propose a particular initialization of the optimization problem that induces an improved convergence rate of alternating gradient descent to recover the matrix $A$. This is supported by both rigorous results and simulations.



**Strengths:**

The paper is clear and well-written. I checked the proofs of the main text and believe they are correct.
I think the paper addresses an interesting problem in matrix factorization, but I am not convinced by the specific initialization that they propose. This is detailed below.

**Weaknesses:**

I would like the authors to discuss more the motivations underlying their paper. My understanding is that factorizing a matrix $A$ in a product $XY^T$ is not hard, for instance one can choose $X=A$ and $Y = Id$. Of course, it is interesting to see if gradient descent is able to recover a factorization of the matrix $A$, given that there is non-convexity and non-smoothness. However, it seems to me that the initialization that you are proposing is similar to the solution $X=A$ and $Y = Id$. In this sense, the papers [YD21] and [JCD22] are stronger as they prove convergence from a Gaussian initialization, that seems further away from the solution.

Of course, a contribution of your paper is to improve the convergence rates. However, it does not seem to me that obtaining optimal convergence rates is an identified challenge in the literature. For instance, you compare your convergence rate to the rate of [JCD22], but again these authors use a more challenging initialization, and prove that incremental dynamics appear, which is much harder than proving a simple convergence rate.

In light of the above points, I think that the comparison with related works in Section 3.1 is a bit unfair.

Given that this model is toyish, it is important to focus on behaviors that we believe to generalize on more sophisticated (matrix multiplication, non-convex) models. While incremental learning is one of them, I didn't understand whether an improved convergence with an atypical initialization satisfies this criteria.

**Questions:**

- Given that the convergence rates are exponential, wouldn't logarithmic y-axes be more suitable to plot the evolution of the errors?
- I think it is a bit tough to skip the proof of Lemma 4.1. The least would be to explain that X -> f(X,Y) is a convex ||Y||^2-smooth function and thus you can use classical (but not really "direct"!) manipulations from convex optimization to derive the bounds.
- Could you comment on why you are using alternating gradient descent rather that full gradient descent? Is it related to using only marginal smoothness in Lemma 4.1? Could your techniques be adapted for full gradient descent?
- l.312: Are you using different Gaussian matrices for X_0 and Y_0?
- l.319: Can you comment on the very small value for \nu?
- Figure 3: It seems that the benefits of the proposed initialization reduces when using large stepsizes. Could you try even larger stepsizes? Can you push simulations to the limit of stability of the gradient descent equations?

Typos
- l.407: "non-asympototic" -> "non-asymptotic"
- l.80: "and" -> "an"
- l.263 and footnote same page: "PL-equality" -> "PL inequality"?
- l.280: "interation" -> "iteration"
- l.300: inconsistent notation for the transpose

**Limitations:**

See weaknesses section.

---

### Official Review · Reviewer_Ewek · 2023-07-06

**Soundness:** 3 good
**Presentation:** 4 excellent
**Contribution:** 4 excellent
**Rating:** 8
**Confidence:** 4

**Summary:**

This paper provides a new analysis for the convergence of alternating gradient descent on the low-rank matrix factorization problem $min_{X,Y} f(X,Y) = ||XY - A||_F^2$. The authors show that, by warm-starting the solution using the target matrix $A$, and appropriate step size scaling, they can achieve $\epsilon$ error in $f$ in $O(\kappa^2 \log \frac{||A||_F}{\epsilon})$ iterations of alternating gradient descent, where $\kappa$ is the pseudo-condition number of $A$, as long as the rank of $XY$ is larger than the rank of $A$ by a constant factor. This improved the best known dependence of $\kappa$ from $\kappa^3$ to $\kappa^2$. Also, in contrast to some previous works, the initialization step does not require computing an SVD of $A$.

**Strengths:**

- The main technical contribution of this paper is the introduction and analysis of an asymmetric warm starting rule for matrix factorization. To the best of my knowledge this is an original technical contribution.
- The theoretical analysis is interesting and advances the state of the art. Matrix factorization is a fundamental optimization task that among others, can be viewed as a subproblem of neural network training. As a result, progress in theoretical understanding of optimization algorithms is highly relevant.
- The experimental results are interesting and show a surprising quality gap between different initialization methods.
- The paper is very well written and technically sound.

**Weaknesses:**

- It seems to me like the assumption in line 454 that $V^\top \Phi_1$ has i.i.d entries is incorrect, and it only has i.i.d columns. In this case, a different lemma should be used in place of Proposition A.1. Does this change anything in the bounds?

- I have some comments about the experiments in Section 6. It seems like the authors compare different algorithms using the same step size, however in my opinion it would be fairer to compare using the best fixed step size for each algorithm. This is especially so because of the special imbalanced step size setup in the authors' algorithm. For example, in Figure 3, Random and ColSpan significantly improve when increasing the step size from $0.001$ to $0.1$. What happens if the step size is increased even more? Can we rule out that the improvement in convergence is because of the step size magnitude?

Some typos:
- decent -> descent in multiple theorem/lemma statements.
- Line 231: it should not be a strict subset

**Questions:**

1. Could the authors give some intuition behind the importance of imbalanced scaling? I understand that it is used to balance the norms of $X$ and $Y$, but the fact that it is so significant in the experiments is somewhat surprising so I am wondering if there is something deeper.

2. Could this approach be used in the noisy setting, i.e. when $A$ is not exactly low rank? What are the difficulties?

---

> ### Comment · Reviewer_Ewek · 2023-08-18
>
> I thank the authors for their response, they have covered my questions.

---

### Official Review · Reviewer_BUrC · 2023-07-06

**Soundness:** 4 excellent
**Presentation:** 4 excellent
**Contribution:** 3 good
**Rating:** 8
**Confidence:** 3

**Summary:**

The authors explore the use of alternating gradient descent (AGD) with a fixed step size for asymmetric matrix factorization. They demonstrate that a finite number of AGD iterations can achieve a high-probability -optimal factorization, even when starting from an asymmetrical random initialization. Empirical evidence supports the significant convergence rate improvements resulting from their proposed initialization. The authors' proof leverages a uniform Polyak-Lojasiewicz (PL) inequality and Lipschitz smoothness constant, providing a simpler approach for analyzing nonconvex low-rank factorization problems.

The findings in this paper challenge the conventional belief that symmetrical randomization is necessary when the underlying data structure allows for symmetry. This novel result, supported by solid theoretical justification, showcases promising improvements and highlights the potential for broader impacts in machine learning and optimization.

**Strengths:**

- Excellent novelty
- Theoretical proof is elegant
- Promising numerical results

**Weaknesses:**

It is worth noting that there are numerous competitors beyond first-order methods in the field to solve an exact rank-r matrix decomposition problem. To further enrich the discussion, it would be valuable if the authors explored potential avenues for generalizing their results. For example, investigating the application of their findings in matrix completion or the singular value decomposition (SVD) of a low-rank matrix with added noise could provide insights into the broader applicability of their approach.

**Questions:**

There are two parts to the initialization. One is the $A\Phi_1$ vs $\Phi_2$, another is $\sqrt{\eta}$ vs $\frac{1}{\sqrt{\eta}}$. I am curious whether  $\sqrt{\eta}$ vs $\frac{1}{\sqrt{\eta}}$ is the major factor for bringing improvements. For example, what are the results if the authors use symmetrical random initialization with $\sqrt{\eta}$ on one side and $\frac{1}{\sqrt{\eta}}$ on another side?

**Limitations:**

I don't foresee any potential negative societal impact resulting from this research.

---

### Official Review · Reviewer_Zy5R · 2023-07-08

**Soundness:** 4 excellent
**Presentation:** 4 excellent
**Contribution:** 4 excellent
**Rating:** 8
**Confidence:** 4

**Summary:**

This paper proves an improved convergence bound on alternating gradient descent for asymmetric matrix factorization problem, i.e. given $A \in R^{m \times n}$, finding $X \in R^{n \times d}$ and $Y \in R^{m \times d}$ that minimize $||XY^{\top} - A||_F^2.$ This paper establishes that if $A$ is rank $r$ for some $r < d,$ (so the optimal solution is 0) then the algorithm converges to a solution with error at most $\varepsilon ||A||_F^2$ in roughly $\frac{d}{(\sqrt{d}-\sqrt{r})^2}\kappa^2 \log \frac{1}{\varepsilon}$ iterations, where $\kappa = \frac{\sigma_1(A)}{\sigma_r(A)}$. Thus, if the problem is constantly over-parameterized, i.e. $d-r = \Omega(r),$ this gives a dimension independent convergence bound.
A crucial piece in their algorithm and analysis is starting from an unconventional asymmetric initialization, where $X$ is initialized as $\frac{1}{\sqrt{\eta}}A\Phi_1$  and $Y$ as $\sqrt{\eta}\Phi_2,$ where $\Phi_1, \Phi_2$ are (normalized) iid Gaussian matrices, and $\eta$ is the step size.
The paper also reports several numerical simulations demonstrating the significant advantage of their initialization.

**Strengths:**

The (asymmetric) matrix factorization problem is both an important problem in itself, and an important test bed for developing techniques to analyze non-convex optimization. I think this paper takes a significant step towards understanding (alternating) gradient descent in this setting. It improves on the previous best known bound for GD which was ~ $\kappa^3.$
I find the improvement due to the asymmetric initialization quite illuminating, and I think it will inspire other advantageous (yet quick) initializations for descent algorithms.

**Weaknesses:**

I don't see any major weaknesses.

**Questions:**

- I think the bound on line 30 is incorrect? there should be a square-root on d and r.

---

### Author Rebuttal · Authors · 2023-08-07

Thank you to all the reviewers for the helpful comments.

Response to Reviewer Zy5R

- [Typo in bound on Line 30:] Thanks! Yes, the bound should be $\frac{d}{(\sqrt{d} - \sqrt{r})^2}$,  not $\frac{d}{d-r}$


Response to Reviewer BUrC

- [Generality of results] The analysis extends beyond matrix factorization but we prefer not to write details as we are in process of writing up follow-up papers.
- [Importance of two parts to the initialization] Thanks for this point.  Both parts of the initialization are important.  See FIGURE 2 in the attached PDF for a comparison of the convergence behavior when both parts are used, compared to when just one or the other part is used. We will add these empirical comparisons to the revised version.

Response to Reviewer Ewek

- [Assumption that $V^T \Phi_1$ has i.i.d entries] This assumption is correct, since we assume $\Phi_1$ has i.i.d. Gaussian entries and $V$ has orthonormal columns, which implies that $V^T \Phi_1$ has i.i.d. Gaussian entries.

- [Comparison of algorithms using larger step-size]
This is a great point.  Please see FIGURE 1 in the attached PDF, where we consider step-size $1$. Larger step-sizes lead to divergence, so step-size 1 is close to the edge of stability of the algorithms.  Observe that while our initialization has a small advantage, the algorithms are all essentially comparable at the edge.  This should not be surprising, since at $\eta=1$, $1/\sqrt{\eta} = \sqrt{\eta}$ and the asymmetric component of the initialization is minimal.  We will include the comparison for $\eta = 1$ in the revised version and add this commentary.  Understanding the theory of gradient descent for matrix factorization at the edge of stability (e.g., for this example, $\eta \approx 1$) is quite challenging, and beyond the scope of this paper.

- [Typos] Thanks! we will fix these.

- [Intuition for importance of imbalanced scaling] Good point.  Our intuition is as follows: with this particular asymmetric initialization, the $X$ updates remain sufficiently small with respect to the scale of the initialization $X_0$, and the $Y$ updates remain sufficiently large with respect to the initial scale $Y_0$, that alternating gradient descent behaves comparably to the \emph{linear} regression problem $\text{minimize}_Y \| A - X_0 Y \|_F^2$ (that is, $X = X_0$ is kept fixed at its initialization).

- [Noisy setting, i.e. when $A$ is not exactly low rank] Empirically, yes, alternating gradient descent with our initialization converges to the best low-rank approximation to $A$ when $A$ is not exactly low-rank. The theoretical difficulty (although not impossibility!) is in carefully extending our Lemmas 4.5 and 4.6, which are no longer invariants of the algorithm beyond the exact low-rank setting.

Response to Reviewer GCxr

- [Significance of initialization] Our initialization is not comparable to $X = A$ and $Y = Id$.  The sizes of the factors are completely different.

- [Comparison to previous papers] While we agree that the papers $[$YD21$]$ and $[$JCD22$]$ prove convergence from a more generic starting point, our initialization has the same cost as a single step of gradient descent, and thus could potentially lead to an equally cost-effective initialization strategy (for matrix factorization and beyond). In this sense, our initialization is closer to the Gaussian initialization of $[$YD21$]$ and $[$JCD22$]$ than previous lines of work on spectral initializations.

- [Optimal convergence rates as identified challenge] This was not an identified challenge in the literature previously.  One contribution of our paper is identifying this challenge.

- [Behaviors that extend to more sophisticated models] Our theory does extend to more sophisticated models. We prefer not to disclose details as we are in process of writing up follow-up papers.

- [logarithmic y-axes] Yes, good point and we can make this change in the revision.

- [Proof of Lemma 4.1] You are right; we will include at least a summary of the proof as you suggest in the revision.

- [Alternating rather that full gradient descent] Yes exactly, the marginal smoothness from alternating gradient descent allows us to prove a descent lemma, Lemma 4.1, which is the crucial first step in the convergence proof.  We did not see how to prove a descent lemma for (non-alternating) gradient descent, but numerics indicate that the two algorithms behave essentially the same.
\item[$X_0$ and $Y_0$:] We are using independent Gaussians for $X_0$ and $Y_0$, but the theory also holds if we use the same Gaussian matrix for $X_0$ and $Y_0$ (with $Y_0$ re-scaled).  We chose to focus on the case where they are different, to more closely compare to the Gaussian random initialization in $[$YD21$]$ and $[$JCD22$]$.

- [small value for $\nu$ in plot] A smaller $\nu$ leads to slightly better concentration of $f_0 = f(X_0, Y_0)$, see Proposition 4.3 part 2.

- [Benefit of the proposed initialization at large stepsizes:] Good point.  Please see the response to Ewek above, and FIGURE 1 in the attached PDF.

---

> ### Comment · Reviewer_GCxr · 2023-08-11
>
> I thank the authors for their rebuttal. I would like to continue the discussion on the some of the points.
> > [Significance of initialization] Our initialization is not comparable to $X = A$ and $Y = Id$. The sizes of the factors are completely different.
>
> I apologize for the mistake. I should have written $X = 1/\sqrt{\eta} A$ and $Y = \sqrt{\eta} Id$. With this modification, my point still holds. This is a trivial solution for the matrix factorization problem. Your initialization is comparable to this solution. As a consequence, doesn't this explain the advantage of your algorithm over [YD21] and [JCD22]?
>
> >[Comparison to previous papers] While we agree that the papers YD21 and JCD22 prove convergence from a more generic starting point, our initialization has the same cost as a single step of gradient descent, and thus could potentially lead to an equally cost-effective initialization strategy (for matrix factorization and beyond). In this sense, our initialization is closer to the Gaussian initialization of YD21 and JCD22 than previous lines of work on spectral initializations.
> [Optimal convergence rates as identified challenge] This was not an identified challenge in the literature previously. One contribution of our paper is identifying this challenge.
> [Behaviors that extend to more sophisticated models] Our theory does extend to more sophisticated models. We prefer not to disclose details as we are in process of writing up follow-up papers.
>
> Here, I would like to discuss a bit more the scientific methodology. You are studying a problem that can trivially be solved, for instance taking $X = 1/\sqrt{\eta} A$ and $Y = \sqrt{\eta} Id$. Why should we study the performance of gradient descent methods, given that there is a trivial solution available?
>
> Of course, I understand that it is sometimes necessary to study toy problems to extrapolate later to more complicated problems. However, here, you refuse to discuss whether your improved initialization generalizes to more complicated problems, where gradient descent methods would be needed in the absence of closed form solutions.
>
> I think I would be more convinced by applications for which gradient descent methods are needed (no trivial solution exists), but still there is a computationally cheap improvement on the initialization that accelerates the convergence rate.
>
> > We are using independent Gaussians for $X_0$ and $Y_0$, but the theory also holds if we use the same Gaussian matrix for $X_0$ and $Y_0$ (with re-scaled). We chose to focus on the case where they are different, to more closely compare to the Gaussian random initialization in YD21 and JCD22.
>
> That works; I was a bit confused with the notation. I would recommend to make it more specific using indices $1$ and $2$.
>
> > [Benefit of the proposed initialization at large stepsizes:] Good point. Please see the response to Ewek above, and FIGURE 1 in the attached PDF.
>
> Thank you.

---

### Decision · Program_Chairs · 2023-09-21

**Decision:**

Accept (spotlight)

**Comment:**

In this paper, the authors studied the classical problem of asymmetric matrix factorization, and came up with an alternating gradient descent (AGD) algorithm that enjoys provable fast global convergence for solving this problem. The rapid convergence is guaranteed even when the algorithm is initialized at asymmetric random initialization, which expands our understanding of the role of initialization for this nonconvex optimization problem. I would encourage the authors to expand their algorithm and theory to accommodate a broader (and more challenging) class of problems like matrix completion and matrix sensing, in order to further enhance the impacts of this paper.